# Lack of pocket money impacts Ethiopian undergraduate health science students learning activities

**Marema Jebessa Kumsa** *, **Bizuayehu Nigatu Lemu, Teklehaimanot Mezgebe Nguse**

Department of Medical Radiologic Technology, School of Medicine, College of Health Sciences, Addis Ababa University, Addis Ababa, Ethiopia

* maaroj@gmail.com

## Abstract

### Background

The cost of university presents various challenges with regards to students' daily learning activities. This is particularly evident in developing countries, where higher education students face acute financial problems that greatly affect their daily educational activities. In Ethiopia, public university students do benefit from governmental cost-sharing programs. Moreover, health sciences students have additional costs during their clinical placements that are above the common expenses for university students.

### Objectives

Authors aim to explore the challenges that undergraduate health sciences students in their clinical year face with limited pocket money, as well as how students perceive these limited funds affecting their learning activities and their ability to meet challenges.

### Methods

This descriptive qualitative study was conducted at the Department of Medical Radiologic Technology, College of Health Sciences, at Addis Ababa University in Ethiopia. Interviews were conducted between January 28, 2019 and February 1, 2019 with twelve students; and only ten participants were included in the study. The semi-structured questions explored participants' experiences and perceptions regarding the challenges of a lack of pocket money and its impacts on their learning activities. Their reaction to financial challenges was assessed.

### Results

Four themes that are related to the impact of a lack of money on learning activities emerged from our interviews. First, students believed that their difficulty in obtaining pocket money from family or other funding sources contributed to their financial stress, which negatively impacted their learning. Moreover, their difficulty in affording the basic needs for a student greatly affected their learning abilities in the classroom as well as in their clinical placements.

**Data Availability Statement:** All relevant data are within the manuscript and its Supporting information files are also attached within.

**Funding:** The author(s) received no specific funding for this work.

**Competing interests:** The authors have declared that no competing interests exist.

**Abbreviations: AAU**, Addis Ababa University; **SES**, Socioeconomic status.

The ability to self-manage was also a significant concern for students, with the pressure to use self-control and proper money management adding to their financial stress. Lastly, students observed that the lack of pocket money affected their ability to make social connections at university, which they saw as negatively impacting their learning abilities.

## Conclusion

Ethiopian undergraduate health sciences students faced many challenges due to the lack of pocket money and these challenges affected student learning both directly or indirectly. Based on our data, we believe that the underlying causes of student financial hardship can be addressed by increasing public awareness of university expenses, clarifying the cost-sharing system to the public, redesigning the cost-sharing policy, and improving university services. Additionally, teaching students self-management skills is also another area that could increase student success.

## Introduction

The United Nations Educational, Scientific and Cultural Organization's World Conference on Higher Education stated that "higher education needs to be a fundamental right for all", regardless of a student's socioeconomic status (SES) [1]. However, students from low SES backgrounds have lower educational aspirations, persistence rates, and educational achievements during college [2–5]. Moreover, several previous studies suggest that during clinical placements, health sciences students face increased financial costs such as transportation, food expenses, clothing, and other material needs [6–8], further exacerbating the stress on low SES students.

In Ethiopia, the Ethiopian Higher Education Proclamation and Cost Sharing Regulations provide room and board, and cover the tuition fees for those public university students who pass the Ethiopian Higher Education Entrance Examination; however, it does not provide students with pocket money for other expenses [9].

The aim of this study is to explore the impact of a lack of pocket money on undergraduate health sciences students' learning activities during their clinical years, with pocket money being defined as the income that a student receives from a parent or guardian [10]. This research helps identify the challenges that undergraduate health sciences students face due to the lack of pocket money and the perceived effects of their financial hardships on their learning activities.

## Methods

### Study design and setting

This study is a descriptive qualitative study that was used to explore the impacts of a lack of pocket money on undergraduate health sciences students' learning activities during their clinical years. Semi-structured interviews were used to gather data and the questions aimed to determine the students' perceptions of the effects of their financial hardships on their learning activities. This study was conducted with third- and fourth-year students from the Department of Medical Radiology Technology, College of Health Sciences, Addis Ababa University (AAU) in Ethiopia.

## Ethical considerations

Ethical clearance was obtained from the Department of Medical Education Ethical Review board. All potential participants were told that they have the right to decline participation in the study and that their non-involvement would not affect their status as health sciences students (S1 Appendix). In addition, they were informed that if they chose to participate in the study it would be completely anonymous. Written informed consent was obtained from each individual participant (Appendix B in S1 File).

## Participant recruitment and data collection

In order to meet the objectives of this study, purposeful sampling was employed to identify financially challenged third- and fourth-year health sciences students. First, we invited those who had submitted letters to the department seeking financial support to participate in the study. When interested individuals arrived, they were informed of the project and asked to declare if they are faced financial challenges. Subsequently, snowball sampling was used to obtain the remaining potential study participants. During all interviews, participants were asked to self-declare their financial status. In the end, seven third-year students and five fourth-year students were interviewed.

Our data starts to be similar after five participants were interviewed. Thematic saturation was reached after reviewing the transcripts of ten participants. Moreover, during our transcription of the participants' interviews, we found that two of the participants self-declared that they were experiencing financial hardship, but from their answers, this was determined not to be the case and they were dropped from the study. Therefore, interviews from ten participants were transcribed verbatim, and translated. The characteristics of the participants are summarized in the Table 1.

## Interviews

The semi-structured interviews were designed using Pierre Bourdieu's concepts of social capital [11]. Two pilot interviews (not included in the analysis) were carried out to help refine the interview questions. One of the pilot interviews was conducted by the corresponding author with another colleague and the other one was conducted by the interviewer. An emphasis was placed on the lack of pocket money and its impacts on learning activities (S2 Appendix).

**Table 1. The characteristics of the ten participants enrolled in this study.**

| Participant Label | Gender | Age | Year of Study | Place of Origin | Parent/family job |
|---|---|---|---|---|---|
| A | Male | 21 | 4 | Addis Ababa | Government |
| B | Male | 23 | 4 | Oromia region | Farmer |
| C | Male | 24 | 4 | Amhara Region | Self-employed |
| D | Male | 23 | 4 | Oromia Region | Farmer |
| E | Female | 22 | 3 | Oromia Region | Government |
| F | Female | 21 | 3 | Addis Ababa | Self-employed |
| G | Male | 21 | 3 | Amhara Region | Teacher |
| I | Male | 20 | 3 | Oromia | Farmer |
| H | Male | 21 | 3 | Addis Ababa | Self-employed |
| J | Female | 20 | 3 | Amhara Region | Teacher |

The interviewer, Debela Gela, was an instructor from the Department of Nursing, AAU. The interviewer had never taught the participants, nor had any role in the Medical Radiologic program, and was fluent in Amharic and Afaan Oromo.

The interviews began on January 28, 2019 and were completed on February 1, 2019. The participants were interviewed in their language of choice (11 in Amharic and one in Afaan Oromo). Interview length ranged between 10 and 25 min. Participants were thanked for their participation and were given a notebook and 50 Ethiopian birr worth of mobile cards each. These gifts were discussed and agreed upon by the researchers.

### Data analysis

Researchers used a qualitative content analysis method [12–14], and thematically analyzed the interviews to explore the challenges that low SES health sciences undergraduate students face in their clinical year. The audio-recordings were transcribed verbatim and translated into English by the researchers. The data were assessed by researchers at multiple levels. The three levels of data analysis (code, category, sub-themes, and theme) were taken as appropriate for coding the data [14]. We held four meetings to transcribe, translate, code, categorize, and theme the data and we coded the transcripts based on our research objectives. The codes, themes, and categories were formulated and modified throughout the entire research process with the full participation of all researchers.

## Results

Four themes emerged from our interviews with health sciences undergraduate students at AAU. These themes included: (1) The challenges in obtaining pocket money, (2) impacts of limited pocket money on essential activities and education, (3) the challenges of self-management, and (4) effects of limited pocket money on socialization. From our data analysis, sub-themes emerged from some of the main themes and categories were merged within some of the sub-themes. Our findings are depicted in Fig 1.

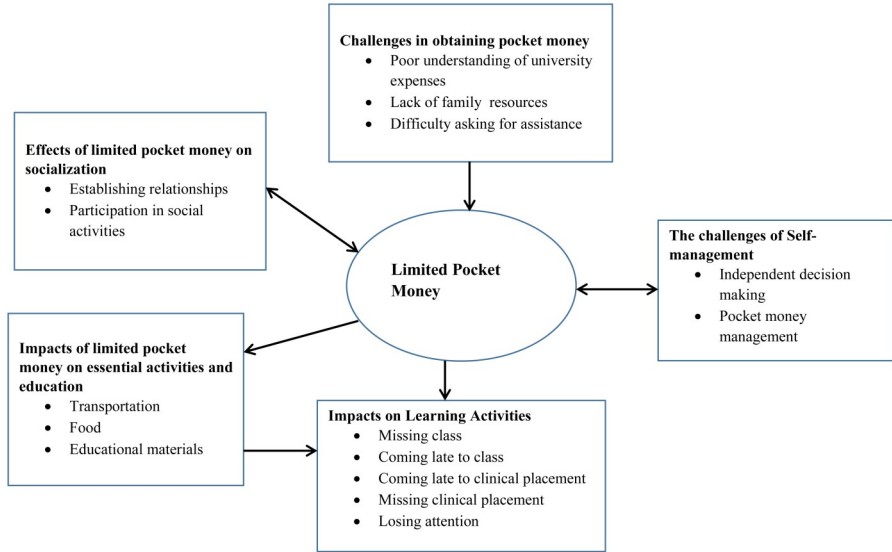

**Fig 1. Conceptual framework for impacts of pocket money on learning activities: Ethiopian undergraduate health science students.**

All participants discussed their experiences related to their lack of pocket money and conceptualized its impacts on their learning process. They also explained the means by which they overcame these problems in order to succeed in their studies. Participant responses were categorized into the aforementioned four themes and Table 2 displays their representative responses.

## Theme 1. Challenges in obtaining pocket money

Nearly all participants mentioned that getting enough money for their various expenses at university was a significant problem. Most students attributed this issue to a lack of awareness of university expenses among parents, absolute poverty, and an inability to request financial aid when needed (Table 2). This lack of pocket money for their daily expenses hampers student learning in a number of ways that will be outlined in subsequent sections.

**Sub-theme 1: Poor understanding of university expenses.** There is a common misunderstanding among parents and students (before they enter university) that once they join public universities, all of their expenses are going to be paid by the government. Parents assume that their child does not suffer from a lack of money and thus do not provide their child with the necessary funds to live comfortably. The belief that there are no expenses for food, transportation, and stationery materials is common. The students identified this as being the most important factor that contributes to their financial trouble.

Additionally, some families do not emphasize the importance of education, and hence, do not give sufficient support to their children. Given this attitude, some parents show a lack of interest in even discussing the financial problems that accompany university life. Participants described such family members as 'education ignorant'. Students from these families not only feel financial pressure, but also feel unsupported, which can be detrimental to their ability to learn.

**Sub-theme 2: Lack of family resources.** Some students explained that even though some of their parents know the expenses that are associated with university life, their economic status does not allow them to cover those expenses. This financial burden is a significant obstacle for those students trying to get the most out of their education. These students are usually advised and taught by their parents how to withstand the financial challenges in order to be successful (this will be discussed further in Theme 3).

**Sub-theme 3: Difficulty in asking for assistance.** Participants explained that given their parent's poor understanding of university expenses, they felt that they must develop the ability to persuade parents and family members to give them pocket money. Unfortunately, several participants did not ask parents or family members for money because of the lack of cultural acceptance for this sort of behavior. Thus, many participants identified asking for pocket money as a real problem and they mentioned that this contributed significantly to them being unable to afford what they need, which had a significant impact on their learning activities.

## Theme 2: Impacts of limited pocket money on essential activities and education

Several study participants indicated that the lack of pocket money had a tremendous impact on their daily lives, particularly affecting their ability to accomplish their learning goals. Most participants stated that the small amount of pocket money that they had went to fulfilling their basic needs, leaving them unable to afford other very important materials and services. This included transportation costs, food expenses, the purchasing educational resources, and other smaller needs, which are discussed below (Table 2).

**Table 2. Representative quotes, within the four identified themes, from health sciences students at Addis Ababa University regarding the effects of a lack of pocket money on learning.**

| Themes | Sub-themes | Verbatim Quotes |
|---|---|---|
| The Challenges in obtaining Pocket Money | *Poor understanding of university expenses* | 'My parents do not know about handouts, assignments, and that I go to my clinical placements using transportation...they do not know and have no experience with higher education, and I am afraid to talk about all of these issues with them.' B |
| | | '...my parents assume that the government covers every expense, and think that there are no additional expenses which the government does not cover. I use an exercise book that was bought by my parents once a year when we were in lower and high schools. However, at university there are handouts which I pay to copy and my parents do not know about handouts, assignments, and that I am going to my clinical practice using transportation...they are not educated and they have no experience with university life. So there are a lot of problems.' B |
| | | '...from my parents' perspective, they thought that there were no expenses in a governmental university and my opinion was the same at that time. We talked about the government covering everything. However, after coming here (AAU) it was evident that this was not the case.' D |
| | | 'My parents have their business; they work at Merkato (the biggest market in the Addis Ababa, Ethiopia). Both are educated up until seventh grade. They do not care about education. It is up to me; I have to learn on my own. They assume that I need nothing.' F |
| | *Lack of family resources* | 'My parents believe that we can be changed by education. However, they do not have and cannot provide me with better things. They give me what they can from what they have.' H |
| | *Difficulty of asking for assistance* | '...you have to talk about the expenses at university and convince your families and others to get the money that you need. However, I lack these persuasion skills...the other possibility is that you have to ask for financial aid...unfortunately there is no well-established financial aid.' I |
| Impacts of limited pocket money on essential Activities and education | *Transportation* | '... when I ran out of pocket money I went on foot. Sometimes I arrived late to class and this made me not as attentive. Other times I missed class completely.' A |
| | | '... when I go to my clinical practice there is no convenient transport service. I remember that I had to go on foot many times. The clinic takes attendance, so I often had no choice but to go on foot.' D |
| | | '...during our first year as health science students there is no place that we need to go...; however, in the second year we start our clinical practice. The transportation service (which is provided by the government) is not suitable or available when we need it. Therefore, we are forced to pay for transportation'. E |
| | *Food* | 'Time goes by fast. For example, the purchasing power of our birr when we were in year one and two is not the same as when I was in year three and four...the price of food goes up' A |
| | | '...for example, I often eat in the cafeteria. However, since the food is not very good, I sometimes eat off campus, which can be expensive.' C |
| | | My expenses increased after I joined AAU. Before I joined, I ate food from home. But, now I eat from the university cafeteria or have to buy food. When I cannot afford food, I do not eat. Thus, I am often hungry and cannot study.' H |
| | *Educational Resources* | 'I cannot access materials on the internet in my dormitory...so I have to wait until I come to another university campus to use the internet. Look, if I had enough money, I would just use my mobile phone's internet access to get on the internet at any time and place.' B |
| | | '...in Tikur Anbessa (one the campus where we had classes), Wi-Fi is available. However, when we go to Sefere Selam (the other campus where we live) there is no internet access and poor access to electricity. So, the university must work on this issue.' C |
| | | '...at our campus there is no good internet connectivity...therefore, I use internet off campus by paying for it.' E |
| | | '...typically, we have to pay out of pocket for the small things like pens, notebooks, copying handouts, and printing assignments.' E |

*(Continued)*

**Table 2.** (Continued)

| Themes | Sub-themes | Verbatim Quotes |
|---|---|---|
| The Challenges of self-management during financial hardships and its effects learning activities | *Independent decision-making* | 'University is a place where a person confronts some of life's greatest challenges. It is where one faces a crossroad: a way to bad and good. So, it is better to control oneself.' A |
| | | '….You have to manage everything. Here, there is no family around to guide you. There are so many students and some smoke and drink alcohol. You have to decide your future for yourself.' B |
| | *Pocket money management* | 'Let me tell you the truth. If I want to have what is necessary for me, the money that I get is not enough. But, I have to compromise some needs to survive.' A |
| | | '… I do not spend for unnecessary things. When I run out of pocket money I borrow from my friends and give them back when I get money from my parents.' B |
| | | '…everything you use comes out of your pocket. You cannot ask your family every day. You are given money for a period of time and must use this money for little and big things. Therefore, this is a challenge.' E |
| | | '…One has to use the small amount of pocket money received sparingly…' I |
| | | '…when I was with my parents, there were very few things I had to worry about. Here, I have to worry about how to manage my money…' J |
| Effects of limited pocket money on socialization | Establishing relationships | '…sharing resources among students is good if a common understanding is created. At university, you have some students that have a lunch and some students that go without eating. Thus, sharing our resources can help alleviate these problems.' F |
| | | '…you have to make good relationships with students if you want to succeed.' G |
| | Participation in social activities | '…here, I need a lot… I need money to go off campus for refreshments with my friends.' C |
| | | '…when I failed to get what I want then I get depressed… For example, when my friends go to the cinema and I can't go, I feel It.' H |
| | | '… I may miss celebrating birthdays with my friends.' J |

**Sub-theme 1: Transportation.** Many participants mentioned that most of their pocket money was spent on transportation. Data collected from the participants revealed that a lack of pocket money affects their capacity to secure convenient transport from where they live to where they learn; they identified that the campus where they live and the campus where they have class are far apart and the transportation that the university provides is usually unreliable. Moreover, students from Addis Ababa who were not entitled to get a dorm room and live with their parents get cash allowances instead of using the university cafeteria and dormitory as part of the cost sharing regulations; however they do not receive any money for transportation. For health sciences students, this problem is compounded by the need to absorb the cost of transportation to their clinical practice away from the university campus. The participants in this study indicated that when students run out of pocket money and are unable to afford transportation, they either arrive late or miss class altogether. The participants also indicated that arriving late negatively impacted their concentration and thereby their ability to learn.

**Sub-theme 2: Food.** Food is one of the biggest expenses for university students. Several issues that were described by participants included an unfavorable timing of food service at the university cafeteria, unpalatable cafeteria food being served, and the need to buy food during times of illness, during clinical placement, and at night. They explained that they typically buy food on campus when they have money, but simply deal with their hunger when they run out of pocket money. These participants indicated that their hunger made it very difficult to study or attend classes.

**Sub-theme 3: Educational resources.** Educational resources are another significant drain on a student's pocket money. The educational resources on which participants spent their pocket money included necessities like stationery materials and internet access.

*Category 1*: *Stationery materials*. Participants indicated that they spent money on buying pens, notebooks, copying and printing handouts and printing assignments almost daily. Several participants identified that their inability to afford educational materials affected their overall learning ability.

*Category 2*: *Access to the internet*. Many of the participants specified that the campus where they live does not provide internet access, and thus they needed to either use mobile data or go to an internet café, both of which require money. Here, the participants identified internet access as an important part of their learning process. Unfortunately, they do not have access to free internet services and are generally unable to pay for it out of pocket.

## Theme 3: The challenges of self-management

Several participants mentioned that leaving their family and living at the university was the most challenging aspect of getting their education. Most of them discussed it as it relates to challenges of self-control and pocket money management, with some participants indicating that these were the keys to success at university (Table 2).

**Sub-theme 1: Independent decision making.**   University is a place where students need to exercise self-control. Most of the participants indicated that when someone does not have enough pocket money and has poor self-control it is very difficult to be successful at university. In addition, they mentioned that good self-control helps to reduce the stress that develops due to a lack of pocket money and therefore improves proper learning.

Many participants focused on self-management and its importance in being successful at university. They believed that, even though many students have financial problems, it is possible to be successful at university if self-management is practiced effectively. Interestingly, most of the participants mentioned that they have developed good self-management skills.

**Sub-theme 2: Pocket money management.**   The effective use of the small amount of pocket money that the students have while in university is very important. Even though some participants mentioned that it is difficult to manage their money, several participants discussed that planning ahead and giving priority to urgent needs is crucial. Those who possessed good pocket money management described that their financial pressures were lessoned by their cautious behavior.

## Theme 4: Effects of limited pocket money on socialization

Several participants believed that success in pursuing a higher education is highly dependent on having a reliable network of friends. They stated that their lack of pocket money affected their ability to make connections with peers and noted that the lack of a social life affected their learning activities. A few participants indicated that meeting with other students and understanding each other's financial status is very helpful in creating a support network. Therefore, a student's inability to go to the cinema, and/or attend ceremonies and other celebrations with their peers often led to students feeling depressed, which negatively affected their learning activities (Table 2).

**Sub-theme 1: Establishing relationships.**   Participants indicated that having compatible friends is of paramount importance to their success at university. They felt that making good friendships with classmates increases one's chances of educational success. Participants also mentioned that leaving their family and entering new environments carries with it many challenges that they believe can be reduced if they have compatible friends. Some mentioned that friends are very helpful when one runs out of pocket money and also when you need help academically. Therefore, having good friends is supportive in times of financial need; unfortunately, they felt that it is difficult to make and maintain friends without having pocket money.

**Sub-theme 2: Participation in social activities.**   A lack of pocket money affects student relationships. The participants of this study identified that they have difficulty either making friends or interacting with existing friends without pocket money. Most of them mentioned that they could not go out with their friends as they could not afford to pay for tea or coffee. Moreover, some participants mentioned that when they were unable to afford refreshments or go out with their friends they became depressed and stressed, which decreased their academic drive and success.

## Discussion

Various factors contribute to student success at university. Here, we investigated how a lack of pocket money influences student learning by discussing the topic with health sciences students in their clinical year at AAU. This study revealed that student financial hardships at AAU occurred mainly due to the lack of understanding about university expenses from families, familial poverty, poor self-management, inhibited social interactions, and poor university administrative services. This study also indicated that the government's cost-sharing system does not fully support financially disadvantaged students. Moreover, we found that the lack of student pocket money negatively impacted student learning both directly (e.g., being unable to afford internet or other materials necessary for learning) and indirectly (e.g., increasing stress and depression in impacted students), and is a substantial concern this student population.

This study showed that the familial habitus of the participants' affected their life at university. Participants described that many of their challenges originated from the significant changes that accompanied moving away from their home and into a university environment. For instance, they went from having their lives mainly managed by their parents and families to self-management. Moreover, they went from living with students with similar lifestyles to living with students of different backgrounds. The ease with which this transition occurs is dependent on the student's social class and related cultural capital, as well as their habitus, which consistently constrains and structures a students' college adaptation. With all these changes, the lack of pocket money adds another stress to their university life.

Several participants mentioned that the absence of a government policy that entitles students to get financial support during their undergraduate studies increased their financial hardship, which is similar to the findings of a study done in the United Kingdom [8]. These financial stresses can be buoyed by a positive social structure. For instance, a previous study demonstrated that students with a low SES and good self-management benefited from gaining social capital [11]. Findings from our study support this idea and indicate that there does indeed appear to be a link between social bonding and achieving relief from financial crisis.

This study also found that students perceived self-management as the most important thing to combat the financial pressures encountered at university. Bourdieu's concept of habitus, describing the way of acting, feeling and being, is related to the participants' idea of self-control and pocket money management.

Similar to a study done in Australia [15], most participants in this study had expected financial hardship to be a substantial concern at the beginning of their degree, however, many of them experienced acute financial hardship during clinical placements. We found that students, who relocated for placements, were economically disadvantaged and experienced increased financial stress [6]. Furthermore, this study indicated that financial stress increased student stress overall, which was also observed in Greece [16] and in another locations in Ethiopia [17].

Unlike the studies presented by Bennett [18] and Trombitas [19], this study does not support the idea that financial pressure causes students to drop-out or quit their education. Instead of dropping out, the participants of the study reported different methods of

overcoming their acute financial problems, including borrowing from friends for urgent needs and developing strong self-management skills (i.e., prioritizing their basic and urgent needs and compromising other non-essentials). Further research is necessary to determine the relationship between financial hardship for Ethiopian health sciences undergraduate students and dropout rates.

The importance of friends and social networks while in university can perhaps be understood with the concept of 'social capital' [20], which is said to be important in communities for overcoming social exclusion. It is used to signify the extent to which people have access to networks, their levels of political and civic engagement, and their membership in various associations [21]. Social capital—or mutual support—seems to be occurring within this study, enabling students to overcome the internal and external problems that they face, including the lack of pocket money. Social relations are shaped largely by the habitus [22], and therefore it is instructive to observe the extent to which the institutional habitus and associated practices can challenge the familial habitus (which Bourdieu shows to be more influential in France).

This study indicated that even though many families misunderstood the expenses associated with higher education and this affected student learning, the students worked diligently to overcome these challenges and often relied upon social circles to help overcome acute financial stress [23].

## Conclusions

The current identified the challenges that undergraduate health sciences students faced due to acute financial hardships by exploring their perceptions and experiences. The main limitation of this study is that it included students from one department and one institution, with small number of participants. However, with this limitation, the current study contributes to a broader overview of the effects of financial hardship on undergraduate academic success and provides a better understanding of its causes. It would be interesting to compare the perceptions and experiences of other students who do not have acute financial problems in the field of study to determine key differences.

The results of this study suggest that policy improvements should occur at the university and national level. With respect to the latter, the Ethiopian cost-sharing policy should be revised to allow for loans or scholarships to students, as well as paid clinical year internships for undergraduate health sciences students. Addressing student financial hardship, especially during clinical placements, is essential for improving the student experience and increasing student competency.

Moreover, creating awareness about student expenses at university within the community will likely improve students' financial status. Additionally, teaching students self-management skills is also another area that could increase student success. Future studies should further investigate the factors that contribute to student financial hardship in the health sciences.

## Supporting information

**S1 Appendix. Information leaflet for participants.**
(DOCX)

**S2 Appendix. Semi-structured interview guide.**
(DOCX)

**S1 File.**
(DOCX)

**S1 Coding. Semi-structured interview coding.**
(DOCX)

## Acknowledgments

Our thanks go to the Department of Health Science Education for giving us the opportunity to conduct this research. We would also like to express our gratitude to Dr. Tina Martimianakis and Dr. Robert Paul for their invaluable guidance, feedback, and constructive comments throughout this project. Finally, we would like to express our appreciation to Debela Gela for his courage in collecting the data.

## Author Contributions

**Conceptualization:** Marema Jebessa Kumsa, Bizuayehu Nigatu Lemu, Teklehaimanot Mezgebe Nguse.

**Data curation:** Marema Jebessa Kumsa.

**Formal analysis:** Marema Jebessa Kumsa.

**Methodology:** Marema Jebessa Kumsa, Bizuayehu Nigatu Lemu, Teklehaimanot Mezgebe Nguse.

**Project administration:** Marema Jebessa Kumsa.

**Supervision:** Marema Jebessa Kumsa, Bizuayehu Nigatu Lemu, Teklehaimanot Mezgebe Nguse.

**Writing – original draft:** Marema Jebessa Kumsa.

**Writing – review & editing:** Marema Jebessa Kumsa, Bizuayehu Nigatu Lemu, Teklehaimanot Mezgebe Nguse.

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
