## [Decision Letter · Decision Letter 0]

6 Aug 2020

PONE-D-19-29025

Title:  Exploring the effects of lack of pocket money on undergraduate clinical year health science students’ learning activities: a Qualitative Study

PLOS ONE

Dear Dr. Kumsa,

Thank you for submitting your manuscript to PLOS ONE. After careful consideration, we feel that it has merit but does not fully meet PLOS ONE’s publication criteria as it currently stands. Therefore, we invite you to submit a revised version of the manuscript that addresses the points raised during the review process.

The reviewers raised significant concerns about several aspects of the paper. Please respond to each concern. In addition, prior to submitting a revision, please have your paper reviewed by an individual skilled in English grammar. There are multiple areas that need to be clarified.

We look forward to receiving your revised manuscript.

Kind regards,

Richard Bruce Mink

Academic Editor

PLOS ONE

Journal Requirements:

2. Please address the following:

- Please modify the title to ensure that it is meeting PLOS’ guidelines (https://journals.plos.org/plosone/s/submission-guidelines#loc-title). In particular, the title should be "specific, descriptive, concise, and comprehensible to readers outside the field" and in this case it is unclear and not well-structured.

- Please justify the small sample size involved in this study with reference to other similar works.

- Please ensure you have thoroughly discussed the potential limitations of this study within the Discussion section.

- Please provide additional details regarding participant consent. In the ethics statement in the Methods and online submission information, please ensure that you have specified what type of consent you obtained (for instance, written or verbal, and if verbal, how it was documented and witnessed).  

Thank you for your attention to these queries.

Reviewers' comments:

Reviewer's Responses to Questions

**Comments to the Author**

1. Is the manuscript technically sound, and do the data support the conclusions?

Reviewer #1: Partly

Reviewer #2: Partly

2. Has the statistical analysis been performed appropriately and rigorously? 

Reviewer #1: Yes

Reviewer #2: I Don't Know

3. Have the authors made all data underlying the findings in their manuscript fully available?

Reviewer #1: Yes

Reviewer #2: No

4. Is the manuscript presented in an intelligible fashion and written in standard English?

Reviewer #1: Yes

Reviewer #2: No

5. Review Comments to the Author

Reviewer #1: Dear authors, thank you for producing this very interesting descriptive study. Your study is a descriptive study based on self-reported challenges in receiving an education due to financial difficulties from 10 health sciences students in Ethiopia. Your study highlights the challenges which Ethiopian health sciences students face due to financial difficulties

The framework within which you conducted the interviews, followed by transcribing, translating, coding and categorizing the data collected, was structured and systematic. The descriptions of what students expressed were interesting and provided granular detail on the challenges they faced.

Some suggestions on improvements:

1. The prose can be edited for clarity. Also edit for grammatical and typographical issues.

2. The conclusion has to be supported by robust data. Consider amending the conclusions to suit the amount of data you have. Eg.To ask the government to adjust certain policies based on interviews of 10 students may not be justified.

3. Add in some benchmark information such as the monthly income of Ethiopians, the school fees, cost of transport etc, so that international readers can understand the context of figures you cited (eg. what is the value 500 Ethiopian Birr).

4. A more in-depth discussion on the significance of each of the Themes and Categories should be done.

5. Consider making a Figure or Diagram to depict your findings graphically

6. Consider clarifying the recruitment process of your interviewees in terms of anonymity, voluntary participation, why they need to be rewarded with mobile cards and money and whether they may be disadvantaged in any way should they not participate. This is important as they may be considered a vulnerable population.

7. Consider increasing the sample size, and also quantifying the challenges that they face in terms of how many respondents face each of the challenges you raised, and also rank these challenges in terms of importance to them. This can be done in subsequent studies.

Reviewer #2: This paper has the potential to make an important contribution to the understanding of the impact of financial limitations on the education of health science students in Ethiopia. It presents qualitative analysis of interviews with students.

However, further clarification of methods is required. In addition, the structure should be revised to enhance the clarity of the manuscript.

Abstract

Background- should include a sentence on previous data and the gap in knowledge

Objectives- tightened up to be more specific

Methods- should start with we conducted 10 semistructured interviews of a description of subjects, time course, place.

Conclusion- should start with the conclusion and then advance to potential solutions. The solutions in abstract should have been discussed in the paper (e.g. educating students on money management)

Body

Introduction

Should include previous data and studies on the challenges that students face and areas of potential misunderstanding (if documented) and then discuss the gap in knowledge

Lines 68-70; is there a reference

Lines 70-71; please explain what cost-sharing is

Line 72; the cited article does not address pocket money

Line 75; the population belongs in the methods

Method

This requires major revision and clearly delineated sections- such as Participants/Recruitment/Ethical considerations, Interviews and then data analysis.

Participants- who was eligible, Recruitment- who were they recruited; Lines 96-96-98 belong in results

Interview-needs more information about the design of interview guide; it should be submitted as a supplement. Pierre Bourdieu's concept of social capital should be described further in that section.

How was interviewer selected. The information about languages used should be in results, as also interview length.

Data analysis- what technology was used for transcription?

Delete lines 115-116 because redundant.

How did the group come to consensus on codes?

Was saturation achieved?

Table 2 should be under results- with elimination of column that includes codes. Headings should be domain, theme, representative quotations.

Consider substituting

"Lack of family resources" for "Poverty"

"Difficulty asking for aid," for "poor skills of asking money

"Transportation" for "transportation expenses"

"Food" for "Food expenses"

It is unclear how the theme of "self-control" or "social life and learning activities" relate to financial limitations

The segment summarizing the themes 127-131 belong in the results

Lines 132-135 are redundant and should be deleted

The section on theoretical approach 136-154 should be deleted

Results

Start with the themes and domains that emerged.

The theme's and subcategories would be better served by capitalizing domains (themes) and using italics for "subcategories" or themes. Because the quotes are already included on the table which will be moved to this section, no need to include again in the narrative. Themes need further detail and elaboration; they are unclear.

Discussion-

The connection to findings and Bourdieu's theory need further clarification. It is difficult to understand. Concepts are introduced in the discussion which were not described in the results (e.g. lines 337-341)

Conclusion-

It is unclear how authors concluded that lack of funds adversely impacted clinical skill acquisition.

Please specify how the Ministry of Science and Higher Education should revise the Ethiopian cost-sharing system.

6. PLOS authors have the option to publish the peer review history of their article (what does this mean?). If published, this will include your full peer review and any attached files.

Reviewer #1: No

Reviewer #2: No

---

## [Author Response · Author response to Decision Letter 0]

22 Sep 2020

Response to reviewers

Subject: Point-by-point response to the reviewers’ comments

First of all I would like to offer my sincere gratitude to academic editor and the reviewers of the at PLOS ONE journal. Please find the point-by-point response provided below.

Response to Reviewer #1: 

The prose can be edited for clarity. Also edit for grammatical and typographical issues. 

• The authors have revised the manuscript several times for grammatical errors, language usage and typographical issues and incorporated all changes. As well professional scientific editing service (Excision editing) has edited the manuscript. All changes are included in the revised version of the manuscript.

The conclusion has to be supported by robust data. Consider amending the conclusions to suit the amount of data you have. Eg.To ask the government to adjust certain policies based on interviews of 10 students may not be justified. 

• We have revised our conclusion of the study. Data presented in the study used to dram conclusion. Example, we added these findings indicate that the department, the school, the college, and the university need to find a way to help students afford university, outside of their room and board. Moreover, creating awareness about student expenses at university within the community will likely improve students’ financial status. Additionally, teaching students self-management skills is also another area that could increase student success. Future studies should further investigate the factors that contribute to student financial hardship in the health sciences.

Add in some benchmark information such as the monthly income of Ethiopians, the school fees, cost of transport etc, so that international readers can understand the context of figures you cited (eg. what is the value 500 Ethiopian Birr). 

• The column indicating average monthly income of students is deleted as there is no established benchmark.

A more in-depth discussion on the significance of each of the Themes and Categories should be done. 

• The result has been modified significantly. We have thoroughly discussed each themes, sub-themes and categories.

Consider making a Figure or Diagram to depict your findings graphically 

• The importance of depicting the findings graphically is considered and we developed a conceptual frame work. The conceptual frame work is added as Figure 1 in the result section.

Consider clarifying the recruitment process of your interviewees in terms of anonymity, voluntary participation, why they need to be rewarded with mobile cards and money and whether they may be disadvantaged in any way should they not participate. This is important as they may be considered a vulnerable population. 

• Participant recruitment process discussed thoroughly in the method section. The rewards are not communicated before the interview and to give after interview is decided by authors. Each participant was informed during consent that participant or not participating will not affect them.

Consider increasing the sample size, and also quantifying the challenges that they face in terms of how many respondents face each of the challenges you raised, and also rank these challenges in terms of importance to them. This can be done in subsequent studies.

• The small size is considered as limitation of this study and we consider other future study.

Response Reviewer #2 

Background- should include a sentence on previous data and the gap in knowledge We added the established data.

• The cost of university presents various challenges with regards to students’ daily learning activities. This is particularly evident in developing countries, where higher education students face acute financial problems that greatly affect their daily educational activities. And gap of study in Ethiopian context is added as: health sciences students have additional costs during their clinical placements that are above the common expenses for university students. 

Objectives- tightened up to be more specific 

• Objectives are modified as: Authors aim to explore the challenges that undergraduate health sciences students in their clinical year face with limited pocket money, as well as how students perceive these limited funds affecting their learning activities and their ability to meet challenges. 

Methods- should start with we conducted 10 semi-structured interviews of a description of subjects, time course, and place. 

• The place and time of study specified in the method. As well the number of participants and tool of data collection discussed.

Conclusion- should start with the conclusion and then advance to potential solutions. The solutions in abstract should have been discussed in the paper (e.g. educating students on money management) 

• We modified the conclusion section and added: Undergraduate students faced many challenges due to the lack of pocket money. Due to challenges students faced affects students learning activities either directly or indirectly. Additionally, teaching students self-management skills is also another area that could increase student success

Should include previous data and studies on the challenges that students face and areas of potential misunderstanding (if documented) and then discuss the gap in knowledge.

• Previous studies on students of low socioeconomic status and the international data have been assessed and the Ethiopian context was analyzed. The absence of study on the matter in Ethiopia is taken as a gap.

Lines 68-70; is there a reference 

• Lines 68-68 are omitted.

Lines 70-71; please explain what cost-sharing is

• Cost sharing in Ethiopia is defined. Ethiopian Higher Education Proclamation and the Cost Sharing Regulations: provide room and board, and covers the tuition fees for those who students who pass Ethiopian Higher education entrance examination and joined public universities, however it does not provide students with pocket money for other expenses.

Line 72; the cited article does not address pocket money 

• The referenced article is corrected to: Bonke J. Do Danish children and young people receive pocket money ? Rockwool Foundat Res Unit. 2013;(57).

Line 75; the population belongs in the methods

• Line 75 is omitted.

Method: This requires major revision and clearly delineated sections- such as Participants/Recruitment/Ethical considerations, Interviews and then data analysis. 

• The method section is revised several times. And subsections are also well organized and enriched. Participants- who was eligible, Recruitment- who were they recruited; Participants selected using purposive sampling technique. Students who show financial support and submitting letter to the department in order to get financial support and those who declare themselves as financially challenged students were eligible.

Lines 96-96-98 belong in results 

• Lines 96-98 have been modified.

How was interviewer selected? The information about languages used should be in results, as also interview length. 

• The interviewer, Debela Gela, was an instructor from the Department of Nursing, AAU. The interviewer had never taught the participants, nor had any role in the Medical Radiologic program, and was fluent in Amharic and Afaan Oromo.

Data analysis- what technology was used for transcription? 

• Authors transcribe manually. Three of the authors transcribed each interview individually and then finally come to consensus after several discussions over transcripts.

Delete lines 115-116 because redundant.

• Lines 11-116 are deleted.

How did the group come to consensus on codes?

• As we with transcription, authors code separately and then came together to discuss and combine the codes. Through several discussions consensus has been reached.

Was saturation achieved? 

• Based on our objective the data became saturated.

Table 2 should be under results- with elimination of column that includes codes. Headings should be domain, theme, representative quotations. 

• Table two moved to under result section.

• The column containing codes is deleted.

Consider substituting

"Lack of family resources" for "Poverty"

"Difficulty asking for aid," for "poor skills of asking money

"Transportation" for "transportation expenses"

"Food" for "Food expenses" 

• We have substituted accordingly.

It is unclear how the theme of "self-control" or "social life and learning activities" relate to financial limitations 

• Participants of the study stated that their lack of pocket money affected their ability to make connections with peers and noted that the lack of a social life affected their learning activities. A few participants indicated that meeting with other students and understanding each other’s financial status is very helpful in creating a support network. Therefore, a student’s inability to go to the cinema, and/or attend ceremonies and other celebrations with their peers often led to students feeling depressed, which negatively affected their learning activities.

The segment summarizing the themes 127-131 belong in the results 

• The section is moved into result section

Lines 132-135 are redundant and should be deleted 

• Lines 132-135 are deleted.

The section on theoretical approach 136-154 should be deleted 

• The theoretical approach section is removed.

Results: Start with the themes and domains that emerged 

• The result section is modified significantly. It is started with major themes and explained. And diagram depicting the findings is developed.

The theme's and subcategories would be better served by capitalizing domains (themes) and using italics for "subcategories" or themes. Because the quotes are already included on the table which will be moved to this section, no need to include again in the narrative. Themes need further detail and elaboration; they are unclear. 

• Themes, sub-themes and categories are edited accordingly. And quotes are no longer used in the narratives.

Discussion-

The connection to findings and Bourdieu's theory need further clarification. It is difficult to understand. Concepts are introduced in the discussions which were not described in the results (e.g. lines 337-341) 

• The discussion section has been modified and enriched. It is edited for misunderstandings and more clarified.

Conclusion-

It is unclear how authors concluded that lack of funds adversely impacted clinical skill acquisition. Please specify how the Ministry of Science and Higher Education should revise the Ethiopian cost-sharing system. 

• Our data show lack of pocket money affects student’s class attendance, clinical placement attendance, and prolonged studying. This believed that students, miss learning different cases on daily basis as they arrive late, stay there while hungering and total absent. Based on participants’ ideas, The Ethiopian cost-sharing policy should be revised to allow for loans or scholarships to students, as well as paid clinical year internships for undergraduate health sciences students. Addressing student financial hardship, especially during clinical placements, is essential for improving the student experience and increasing student competency.

---

## [Decision Letter · Decision Letter 1]

22 Oct 2020

PONE-D-19-29025R1

Lack of pocket money impacts on learning activities: Ethiopian undergraduate health science students

PLOS ONE

Dear Dr. Kumsa,

Thank you for submitting your manuscript to PLOS ONE. After careful consideration, we feel that it has merit but does not fully meet PLOS ONE’s publication criteria as it currently stands. Therefore, we invite you to submit a revised version of the manuscript that addresses the points raised during the review process.

The authors should be commended for providing this revised manuscript. It is much easier to read and the data are presented with better clarity. However, as outlined by reviewer #2, there are still a few points that need to be addressed that will further improve the paper.

Although the investigators had professional assistance in revising the paper, there are a few areas where the punctuation is incorrect. Specifically, when "however" is used as a conjunction, the proper punctuation is  "first sentence; however, second sentence." In addition, in figure 1, I don't understand why there are five boxes interacting with pocket money when four themes were identified.

We look forward to receiving your revised manuscript.

Kind regards,

Richard Bruce Mink

Academic Editor

PLOS ONE

Reviewers' comments:

Reviewer's Responses to Questions

**Comments to the Author**

1. If the authors have adequately addressed your comments raised in a previous round of review and you feel that this manuscript is now acceptable for publication, you may indicate that here to bypass the “Comments to the Author” section, enter your conflict of interest statement in the “Confidential to Editor” section, and submit your "Accept" recommendation.

Reviewer #1: All comments have been addressed

Reviewer #2: (No Response)

2. Is the manuscript technically sound, and do the data support the conclusions?

Reviewer #1: Yes

Reviewer #2: Yes

3. Has the statistical analysis been performed appropriately and rigorously? 

Reviewer #1: Yes

Reviewer #2: Yes

4. Have the authors made all data underlying the findings in their manuscript fully available?

Reviewer #1: Yes

Reviewer #2: Yes

5. Is the manuscript presented in an intelligible fashion and written in standard English?

Reviewer #1: Yes

Reviewer #2: Yes

6. Review Comments to the Author

Reviewer #1: Dear Authors, thank you for your efforts in replying to our comments and in improving the manuscript. You have produced an interesting article that will give insights to how financial difficulties affect undergraduate health sciences students in Ethiopia.

Reviewer #2: This is MUCH better! It is understandable. Now just needs some refinement, especially the discussion.

1. The discussion needs further organization.

Paragraph- 1 286-292 rehashes what was performed. Key findings (e.g. summarized in the conclusion) should be emphasized and should be the structure upon which each paragraph is based. What is most interesting about the manuscript is that it demonstrates that financial hardships are common and adversely impact many aspects of the students’ education and experience. In addition, the authors identify causes leading to potential solutions. The authors should capitalize on these in their discussion.

The application of Bordeau’s theory remains difficult to understand.

2. Conclusions- should summarize implications and next steps

3. Table 1- the participant labels should be revised so they are alphabetical, with subsequent changes in Table 2.

4. Results p 8- some of the theme titles need further refinement, suggested below. Changes made throughout subsequent narrative, tables, figures.

Four themes on the impact of limited pocket money

(1) The challenges obtaining pocket money

3. Difficulty asking for assistance

(2) impacts of limited pocket money on essential activities and education

(3) the challenges of self-management

1. Independent decision-making

(4) effects of limited pocket money on socialization

1. Establishing relationships

2. Participation in social activities

5. Figure 1 p 8 This figure outlining the conceptual framework was an excellent addition. However, the directionality of the relationships (arrows) were unclear. Is central circle/spoke- “limited pocket money”? For some it aligns with themes, for others it doesn’t

Minor

Page 15, line 171- the belief that there are no expenses for food, transportation and writing materials is common.

Page 16- lines 207-210 revise.

Results- Do not reiterate (Table 2) in each subtheme. Introduced in results- may introduce in themes,

Page 6, lines 106-108- Thematic saturation was reached after reviewing the transcripts for 10 participants.

7. PLOS authors have the option to publish the peer review history of their article (what does this mean?). If published, this will include your full peer review and any attached files.

Reviewer #1: No

Reviewer #2: No

---

## [Author Response · Author response to Decision Letter 1]

4 Nov 2020

Rebuttal letter

To academic editor:

 Figure 1. There are four themes emerged from the data. In the conceptual framework we tried to show the relationship between limited pocket money and learning activities. Some of the themes emerged are factors leading to limited pocket money, some are emerged due to lack of pocket money and others are voce-versa. For instance, poor understanding of university expenses, lack of family resources and difficulty of asking for aid are the main factors causing lack of pocket money. While, maintaining social life and self-management could lead to limited pocket money and limited pocket money also lead to poor social life activities and poor self-management. On the other hand, limited pocket money leads unable to afford very important needs. Finally, the figure shows that limited pocket money and inability of affording very important needs Impacts on Learning Activities. Learning activities are affected by missing class, coming late to class, coming late to clinical, placement, missing clinical placement and loosing attention.

We correct punctuation whenever however is used.

To reviewer 2

The central circle was pocket money and changed to limited pocket money. The directions show the relationship between limited pocket money, emerged themes and its impacts on learning. In some the limited pocket money has unilateral direction and in others bilateral and vice-versa.

Table 1 and 2 are modified so that the participants’ orders are listed alphabetically in both tables.

The discussion part is modified accordingly. The main findings are emphasized. And the implications and future steps are capitalized in the conclusion.

We found the suggestion for improvement of theme titles and modified accordingly.

Page 15, line 171 is changed accordingly

Page 6, lines 107-108 is modified to thematic saturation is reached after reviewing the transcripts for ten participants.

The first paragraph of the discussion part is modified. We tried to focus on the main findings in discussion followed.

The conclusion summarized with implications and future steps in solving the raised in the paper.

Referencing (table 2) redundantly in subthemes is correct, so table 2 is referenced in the result section and main themes.

---

## [Editor Report · Decision Letter 2]

12 Nov 2020

PONE-D-19-29025R2

Lack of pocket money impacts on learning activities: Ethiopian undergraduate health science students

PLOS ONE

Dear Dr. Kumsa,

Thank you for submitting your manuscript to PLOS ONE. After careful consideration, we feel that it has merit but does not fully meet PLOS ONE’s publication criteria as it currently stands. Therefore, we invite you to submit a revised version of the manuscript that addresses the points raised during the review process.

While this revision has addressed the reviewers' concerns and the paper reads much better, there are a few more corrections needed. These relate to sentence construction and appropriate punctuation and include:

Abstract

page 3, lines 50-51: change sentence to "The challenges students faced affected their learning activities, either directly or indirectly."

page 3, line 55: add a period at the end of the sentence

Introduction

page 4, line 74: correct punctuation- "...examination and joined public universities; however, it does not...."

Methods

page 6, lines 107-108: change sentence to "...we realized that thematic saturation was reached after reviewing the transcripts of ten participants."

page 7, line 137: delete the comma after process

Results

page 15, lines 164-165: delete bold

page 15, line 182: correct point size of "how."

Discussion

page 22, line 318: add comma: "...beginning of their degree, however, many of them...."

page 23: lines 342-343: Rewrite this sentence because I do not understand it: "This is similar to a previous study title on Causes and Consequences of Higher Education:"

I suggest that you have an expert in English review the paper before you submit the revision.

We look forward to receiving your revised manuscript.

Kind regards,

Richard Bruce Mink

Academic Editor

PLOS ONE

---

## [Author Response · Author response to Decision Letter 2]

20 Nov 2020

Response to Academic Editor

• Om page 3 we changed the sentence and modified it.

• We added a period on page three at the end of the sentence.

• On page 4 we corrected the punctuation accordingly.

• On page 6 we changed the sentence to “Our data starts to be similar after five participants were interviewed. Thematic saturation was reached after reviewing the transcripts of ten participants.”

• On page 7, comma is deleted after process.

• On page 15, the bold is deleted.

• On page 15, the pint size of how is corrected.

• On page 22, comma has been added.

• On page 23, the confusing sentence was deleted.

The titled is modified based on professional editor comment. Therefore the tile is modified from; Lack of pocket money impacts on learning activities: Ethiopian undergraduate health science students to Lack of pocket money impacts Ethiopian undergraduate health science students learning activities

Table 1 is formatted by professional editor and accepted.

---

## [Editor Report · Decision Letter 3]

25 Nov 2020

Lack of pocket money impacts Ethiopian undergraduate health science students learning activities

PONE-D-19-29025R3

Dear Dr. Kumsa,

We’re pleased to inform you that your manuscript has been judged scientifically suitable for publication and will be formally accepted for publication once it meets all outstanding technical requirements. Thank you for having a professional editor review the manuscript.

Kind regards,

Richard Bruce Mink

Academic Editor

PLOS ONE
---

## [Editor Report · Acceptance letter]

27 Nov 2020

PONE-D-19-29025R3 

Lack of pocket money impacts Ethiopian undergraduate health science students learning activities 

Dear Dr. Kumsa:

I'm pleased to inform you that your manuscript has been deemed suitable for publication in PLOS ONE. Congratulations! Your manuscript is now with our production department. 

Kind regards, 

on behalf of

Dr. Richard Bruce Mink 

Academic Editor

PLOS ONE